**Review**

data science; deep learning; feature extraction; image recognition; machine learning.

**Corresponding authors:**
Han Han;
Email: hhan@njfu.edu.cn
Yun Zhou;
Email: zhouyun@purdue.edu

**Associate Editor:** Dr. Ross Sozzani

# The pipelines of deep learning-based plant image processing

Kaiyue Hong[1], Yun Zhou[2] and Han Han[1]

[1]Co-Innovation Center for Sustainable Forestry in Southern China, College of Life Sciences, Nanjing Forestry University, Nanjing, China; [2]Department of Botany and Plant Pathology, Center for Plant Biology, Purdue University, West Lafayette, IN, USA

## Abstract

Recent advancements in data science and artificial intelligence have significantly transformed plant sciences, particularly through the integration of image recognition and deep learning technologies. These innovations have profoundly impacted various aspects of plant research, including species identification, disease detection, cellular signaling analysis, and growth monitoring. This review summarizes the latest computational tools and methodologies used in these areas. We emphasize the importance of data acquisition and preprocessing, discussing techniques such as high-resolution imaging and unmanned aerial vehicle (UAV) photography, along with image enhancement methods like cropping and scaling. Additionally, we review feature extraction techniques like colour histograms and texture analysis, which are essential for plant identification and health assessment. Finally, we discuss emerging trends, challenges, and future directions, offering insights into the applications of these technologies in advancing plant science research and practical implementations.

## 1. Introduction

In the digital age, large-scale plant image datasets are essential for advancing plant science, yet their efficient processing remains challenging; artificial intelligence (AI) and deep learning (DL) offer transformative solutions by enabling machines to simulate human intelligence in tasks like image recognition and decision-making (Williamson et al., 2023). In plant research, machine learning (ML), a subset of AI, enables automatic plant image analysis, allowing computers to learn and improve without explicit programming by identifying patterns in data to predict outcomes and make decisions through supervised, unsupervised, and reinforcement learning approaches (Silva et al., 2019). DL, a specialized branch of ML, uses multi-layer neural networks to process complex plant image data, automatically extract features, and perform tasks like classification and prediction, driving significant advancements in plant image analysis, especially in plant growth monitoring and disease detection (Saleem et al., 2019). Together, AI, ML, and DL propel innovation in plant science – ML enables learning from data, while DL leverages deep neural networks for advanced image analysis – driving transformative progress in plant research.

Processing and analyzing high-resolution plant images pose challenges for image processing algorithms due to plant diversity in colour, shape, and size, with additional complications from complex backgrounds and dense leaf structures affecting segmentation and feature extraction (Sachar & Kumar, 2021). To tackle these challenges, tailored methods in preprocessing, feature extraction, and data augmentation have been developed, showing strong effectiveness in plant image processing (Barbedo, 2016). For example, data augmentation methods like random rotation and flipping improve model adaptability to plant diversity by helping it learn more robust features (Cap et al., 2020). Furthermore, targeted approaches like colour normalization and background suppression improve feature recognition accuracy, reduce external interference, and highlight plants' distinct visual characteristics, optimizing workflows and enhancing plant image analysis accuracy and efficiency (Petrellis, 2019).

Challenges like data acquisition and the lack of high-quality annotated data hinder the widespread adoption of DL technologies in plant science. Image recognition, the first and

crucial step in plant image processing, has been significantly advanced by the rapid development of DL, particularly Convolutional Neural Networks (CNNs) (Cai et al., 2023). CNNs are a type of feedforward neural network and a representative algorithm of DL, using convolutional calculations and possessing a deep structure (Kuo et al., 2019). The performance of CNNs in plant species recognition has been thoroughly evaluated on several large public wood image datasets, consistently demonstrating high accuracy. For example, CNN models achieved 97.3% accuracy on the Brazilian wood image database (Universidade Federal do Paraná, UFPR) and 96.4% on the Xylarium Digital Database (XDD), clearly outperforming traditional feature engineering methods. CNNs are both effective and generalizable for wood image recognition tasks (Hwang & Sugiyama, 2021).

Large language models (LLMs), such as ChatGPT, are advanced DL models that, when combined with domain-specific tools like the Agronomic Nucleotide Transformer (AgroNT) – a novel DNA-focused LLM – have demonstrated great potential in plant genetics and stress response studies (Mendoza-Revilla et al., 2024). By analyzing the genomes of 48 crop species and processing over 10 million cassava mutations, the LLM-based tools offer valuable insights into plant development, interactions, and traits, advancing gene expression profiling and opening new research possibilities (Agathokleousb et al., 2024). Notably, LLMs have revealed new insights by uncovering non-obvious regulatory patterns in promoter regions, predicting the functional impacts of non-coding variants, and suggesting novel gene-stress associations that were previously unrecognized using traditional bioinformatics approaches (Mendoza-Revilla et al., 2024). For example, AgroNT has been shown to predict transcription factor binding affinities across diverse plant species with unprecedented accuracy, enabling the discovery of conserved stress-responsive elements in divergent genomes (Wang et al., 2025). Although still in its early stages, the application of language models in plant biology holds great potential to transform the field, despite currently lagging behind advancements in other domains.

This review evaluates key technologies in plant image processing, such as data acquisition, preprocessing, feature extraction, and model training, examining their effectiveness, limitations, and potential to advance plant science research. It also compares various methodologies and ML models, highlighting their advantages, limitations, and challenges, providing a detailed framework to help researchers make informed decisions in plant image processing studies and applications.

## 2. Data acquisition and preprocessing

### 2.1. Data acquisition and plant feature extraction

Data acquisition and preprocessing are vital for ML in image processing, with high-resolution imaging, unmanned aerial vehicle (UAV) photography, and 3D scanning providing detailed morphological data for DL model foundation (Shahi et al., 2022) (Table 1). While high-resolution devices offer superior quality, regular cameras and smartphones provide greater accessibility and scalability, enabling large-scale data collection and enhancing dataset diversity and model robustness.

Feature extraction in plant image analysis integrates morphology, physiology, genetics, and ecology, starting with colour features (e.g., histograms, coherence vectors) and followed by morphological features (e.g., area, perimeter, shape descriptors) to identify plant traits (Mahajan et al., 2021). Texture features, capturing local variations in images, are crucial for species differentiation and disease detection, revealing surface structures like roughness and contrast (Mohan & Peeples, 2024). CNNs have proven effective in managing complex plant images, enhancing the classification and detection of diseases through robust feature extraction methods (Ahmad et al., 2022). Additionally, structural features, such as leaf morphology and spatial arrangements, are extracted using techniques like edge detection and shape description (Shoaib et al., 2023). Lastly, physiological features, including leaf count, size, and vein structure, provide valuable data on plant health and growth dynamics (Bühler et al., 2015). These features can be obtained manually or automatically through image processing, with recent studies favoring automated methods like segmentation and morphological analysis for high-throughput, objective phenotyping.

**Table 1.** Data acquisition techniques and their applications in plant sciences

| Data acquisition techniques | Description | Type of camera | Type of platform | Type of applications | References |
| --- | --- | --- | --- | --- | --- |
| High-resolution imaging | Using high-resolution cameras to capture detailed images of plants. | Visual | Indoor | Plant growth dynamics, plant diseases diagnosis | Duncan et al. (2022) |
| UAV photography | Drone aerial photography provides detailed insights into plant community distribution and condition. | Visual | UAV | Monitoring crop growth in fields, assessing vegetation coverages, and observing environmental changes. | Wu et al. (2022) |
| 3D scanning technology | Capturing plant spatial structure allows the creation of detailed 3D models of plant morphology. | Structured Light Scanner | Indoor | Plant morphology and growth | Nguyen et al. (2016) |
| Light detection and ranging (LiDAR) | Utilizing LiDAR to capture detailed spatial and structural data of plants. | LiDAR | Outdoor | Investigating plant morphology and analyzing their growth patterns. | Forero et al. (2022) |
| Spectral imaging technology | Capturing plant images in specific wavelengths to gather critical information about the health status of plants. | Multispectral | Indoor & Outdoor | Analyzing the photosynthetic efficiency, water content, and nutritional status of plants. | Zhang et al. (2022) |
| Public databases and resources | Collection of large-scale image datasets from various platforms that are accessible for research on plant species, health, and environmental impact. | Multi-platform | | Expanding research datasets for species recognition and ecological studies. | Mano et al. (2009) |

A key advantage of CNNs is their ability to learn hierarchical features from raw images, eliminating manual feature engineering and enhancing model adaptability and performance in diverse plant phenotyping tasks.

## 2.2. Preprocessing techniques

Data preprocessing in plant image analysis includes key steps like cropping, resizing, enhancing, augmenting, and annotating to optimize images for ML models. Cropping and resizing standardize dimensions, enhancing computational efficiency and reducing model complexity (Maraveas, 2024). Data augmentation modifies original images to generate new datasets, with techniques like contrast adjustment, denoising, and sharpening enhancing detail visibility and accuracy (Abebe et al., 2023), while augmentation strategies like rotation and flipping diversify the dataset to prevent overfitting and improve model generalization (Syarovy et al., 2024). Despite their benefits, preprocessing steps can cause information loss, requiring a balance between simplifying the model and preserving critical information. While most preprocessing is not labor-intensive, annotating and labeling training data remains highly labor-intensive, often becoming bottlenecks that hinder project progress. Accurate annotation, often referred to as 'ground truth', is essential for supervised learning, as it provides a reliable benchmark for model training and evaluation. In both research and practical applications – such as image recognition, natural language processing, and predictive analytics – the quality of labelled data directly influences model accuracy and reliability (Zhou et al., 2018a, 2018b).

To achieve optimal results, recommended sizes of datasets vary by task complexity. For binary classification, 1,000 to 2,000 images per class are typically sufficient (Singh et al., 2020). Multi-class classification requires 500 to 1,000 images per class, with higher requirements as the number of classes increases (Mühlenstädt & Frtunikj, 2024). More complex tasks, such as object detection, demand larger datasets, often up to 5,000 images per object (Cai et al., 2022). DL models like CNNs generally need 10,000 to 50,000 images, with larger models requiring 100,000+ images (Greydanus & Kobak, 2020). Data augmentation can multiply dataset size by 2–5 times (Shorten & Khoshgoftaar, 2019). Additionally, Transfer Learning, a machine-learning model, is effective for smaller datasets, requiring as few as 100 to 200 images per class for successful training (Zhu et al., 2021).

## 2.3. Commonly used public dataset

The Plant Village dataset is a widely used public resource for DL-based plant disease diagnosis research (Mohameth et al., 2020). It serves as a valuable tool in agricultural and plant disease research, offering a comprehensive collection of labelled images essential for developing and testing ML models for plant health monitoring (Pandey et al., 2024). Its accessibility, diversity, and standardized format make it a benchmark for algorithm development in precision agriculture, contributing to early disease detection and yield management, and addressing global challenges like food security and sustainable farming (Majji & Kumaravelan, 2021). Promoting the use of such datasets can enhance collaboration among researchers, standardize methodologies, and support scalable solutions across various agricultural environments (Ahmad et al., 2021).

Similar to the plant village dataset, other plant image datasets include the plant doc dataset, which contains images from various plant species for plant disease diagnosis (Singh et al., 2020). The crop disease dataset features images of diseases in multiple crops, making it suitable for training DL models, especially for crop disease classification (Yuan et al., 2022). The tomato leaf disease dataset focuses on disease images specific to tomato leaves, supporting research in tomato disease recognition and detection (Ahmad et al., 2020). These datasets are widely used in agriculture, particularly for plant disease detection, crop growth studies, and plant health management, driving the ongoing development of intelligent agricultural technologies.

## 3. Model development and training

### 3.1. The selection of ML model

Image classification, used to categorize input images into predefined groups, is commonly applied in plant identification and disease diagnosis. CNNs, with their strong hierarchical feature extraction abilities, excel in these tasks. Models like AlexNet and ResNet are frequently used to classify plant species and developmental stages (Zhu et al., 2018; Malagol et al., 2025). ResNet, by incorporating residual learning and skip connections, addresses gradient vanishing and degradation in deep networks. Its enhanced model has been applied in high-throughput quantification of grape leaf trichomes, supporting phenotypic analysis and disease resistance studies. CNN-based models generally achieve over 90% accuracy on public datasets, validating their 'High' performance in comparative evaluations (Yu et al., 2021; Yao et al., 2024).

Simpler models like K-nearest neighbors (K-NN) and support vector machines (SVMs) are ideal for smaller datasets with less complex features. Though computationally efficient and easy to implement, they are more sensitive to noise and tend to perform less effectively on complex image data (Ghosh et al., 2022). K-NN, for instance, classifies samples based on proximity in feature space and can be enhanced using surrogate loss training (Picek et al., 2022). SVMs utilize kernel functions like the Radial Basis Function (RBF) to handle non-linear data and prevent overfitting (Sharma et al., 2024). Both are typically rated as 'Medium' in performance due to their limitations in handling large-scale, high-dimensional data (Azlah et al., 2019).

For object detection tasks, which require both classification and localization, models like Faster R-CNN (FRCNN) and You Only Look Once version 5 (YOLOv5) offer high spatial accuracy. FRCNN uses a region proposal network (RPN) and shared convolutional layers to efficiently predict object categories and bounding boxes (Deepika & Arthi, 2022). YOLOv5 enables real-time detection and has been applied to UAV-based monitoring for early detection of pine wilt disease (Yu et al., 2021). In dense slash pine forests, improved FRCNN achieved 95.26% accuracy and an $R^2$ of 0.95 in crown detection, showcasing the utility of deep learning for woody plant monitoring (Cai et al., 2023).

Other advanced models include deep belief networks (DBNs), which use stacked restricted Boltzmann machines (RBMs) for unsupervised hierarchical learning and are fine-tuned via backpropagation (Lu et al., 2022). Recurrent neural networks (RNNs), particularly long short-term memory (LSTM) networks, are effective for modeling temporal dependencies, such as plant growth simulation using time-lapse imagery (Xing et al., 2023; Liu et al., 2024a, 2024b). Graph neural networks (GNNs) are increasingly used for modeling complex relationships in plant stress response and gene regulation, although they require significant training effort and parameter tuning (Chang et al., 2024).

**Table 2.** A comparison of common ML models in plant recognition and classification

| ML models | Task | Advantages | Disadvantages | Applications | Performance score | References |
|---|---|---|---|---|---|---|
| CNN | Image classification | Highly accurate and suitable for complex image recognition. | Requires a large amount of data with high computational cost. | Recognizing and categorizing complex plant images | High | Yu et al. (2021) |
| SVM | Classification of high-dimensional data | Effectively handles high-dimensional data. | Sensitive to parameter selection and long training times. | Classifying small to medium-sized plant datasets. | Medium | Ghosh et al. (2022) |
| RFs | Multi-class classification | Suitable for multi-class classification tasks and demonstrate robust performance. | The model has high complexity, resulting in longer training times. | Handling complex feature plant classification problems. | High | Pandey and Vir (2024) |
| K-NN | Instance-based classification | Simple and easy to implement, suitable for small datasets. | Sensitive to noise; classification performance depends on distance selection. | Analyzing small-scale datasets or performing preliminary plant recognition. | Medium | Azlah et al. (2019) |
| DBNs | Deep feature learning | Offers powerful representation capabilities through deep feature learning. | Training is complex and requires adjustment of multiple hyperparameters. | Deep-level plant feature learning and classification | Medium | Shoaib et al. (2023) |
| Transfer Learning | Transfer knowledge from pre-trained models | Using pre-trained models reduces the amount of training data required. | Adequate pre-trained models related to the task are required. | Quickly applicable to new plant classification tasks | Medium to High | Shahoveisi et al. (2023) |

In scenarios with limited labelled data or domain shifts, transfer learning is particularly valuable. By leveraging pre-trained models, it enables medium to high performance in plant classification and disease recognition tasks (Wu et al., 2022). Advanced architectures like GoogLeNet, with its multi-scale inception module, further enhance classification accuracy. For instance, a GoogLeNet model achieved F-scores of 0.9988 and 0.9982 in classifying broadleaf and coniferous tree species, respectively, after 100 training epochs (Minowa et al., 2022).

In summary, each model class exhibits distinct strengths. CNNs specialize in image classification; SVMs and K-NN are optimal for simpler datasets; FRCNN and YOLOv5 excel in object detection; DBNs and RNNs support hierarchical and temporal modeling; GNNs tackle high-dimensional interactions; and transfer learning enhances adaptability across domains. The qualitative performance ratings (High/Medium) presented in Table 2 synthesize evaluation metrics like accuracy, precision, recall, and F1-score across representative studies, enabling researchers to select appropriate models based on task complexity and dataset characteristics.

### 3.2. The integration of language models

In recent years, with the widespread application of LLMs such as BERT and the GPT series in natural language processing and cross-modal learning, plant science research has also begun exploring the integration of LLMs into areas such as gene function prediction, literature-based knowledge mining, and bioinformatic inference. For instance, LLMs can automatically extract potential functional annotation information from a large body of plant gene literature, aiding in the construction of plant gene regulatory networks. Moreover, due to their powerful contextual understanding capabilities, LLMs demonstrate enhanced accuracy and generalizability in predicting gene expression patterns across species (Zhang et al., 2024). The application of LLMs in plant biology is beginning to transform the field by driving advancements in chemical mapping, genetic research, and disease diagnostics (Eftekhari et al., 2024). For instance, by analyzing data from over 2,500 publications, researchers have revealed the phylogenetic distribution of plant compounds and enabled the creation of systematic chemical maps with improved automation and accuracy (Busta et al., 2024). LLMs and protein language models (PLMs) also enhance the analysis of nucleic acid and protein sequences, advancing genetic improvements and supporting sustainable agricultural systems (Liu et al., 2024a, 2024b). In disease diagnostics, models like contrastive language image pre-training (CLIP) utilize high-quality images and textual annotations to improve classification accuracy for plant diseases, achieving significant precision gains on datasets such as plant village and field plant (Eftekhari et al., 2024).

Similarly, a CNN-based system combining InceptionV3 with GPT-3.5 Turbo achieved 99.85% training accuracy and 88.75% validation accuracy in detecting tomato diseases, providing practical treatment recommendations (Madaki et al., 2024). The feature fusion contrastive language-image pre-training (FF-CLIP) model further enhances this approach by integrating visual and textual data to identify complex disease textures, achieving a 33.38% improvement in Top-1 accuracy for unseen data in zero-shot plant disease identification (Liaw et al., 2025). These advancements highlight the transformative potential of language models in advancing plant biology research and driving sustainable agricultural innovation.

In summary, two notable new insights brought by LLMs in plant science include: (1) the ability to identify and summarize 'potential regulatory information within non-coding sequences', which is often overlooked by traditional models; and (2) the promotion of holistic modeling of plant trait complexity through multimodal integration – such as combining sequence, image, and textual data – offering new avenues for complex trait prediction and breeding design.

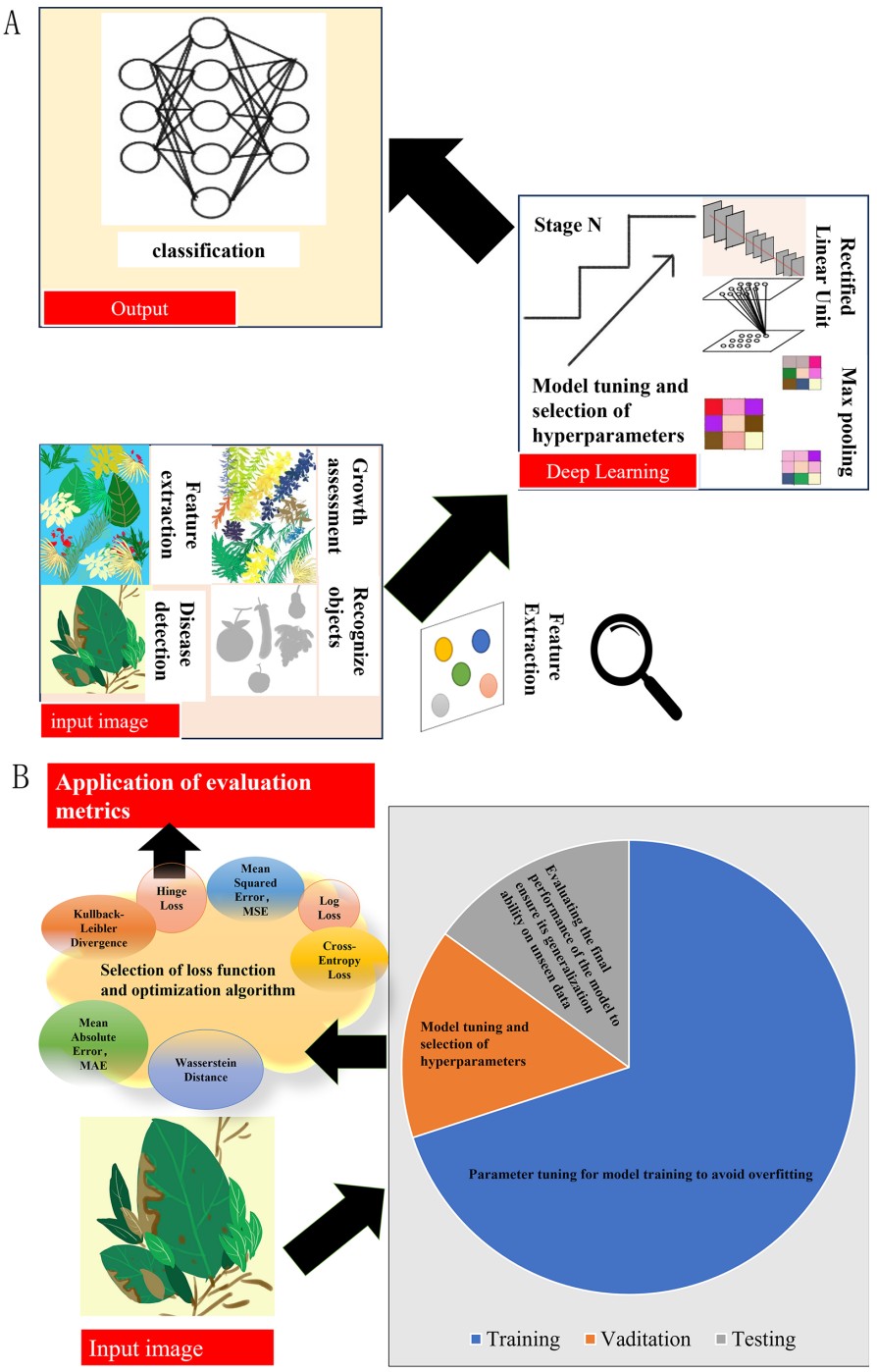

**Figure 1.** CNN and data training process flowchart. A. DL-based image processing flowchart; B. Data training process elowchart.

### 3.3. Model training and evaluation methods

Effective ML model training relies on three key components: data partitioning, loss functions, and optimization strategies. Properly splitting the data (commonly 70:15:15 for training, validation, and testing) ensures good generalization (Figure 1). The training set fits model parameters, the validation set guides hyperparameter tuning, and the test set evaluates final performance, enhancing model robustness(Ghazi et al., 2017). For instance, Mohanty et al. used the PlantVillage dataset (54,306 images), applied an 80:10:10 split and cross-entropy loss with SGD optimization, achieving over 99% accuracy and demonstrating the efficiency of deep learning

in plant disease identification (Mohanty et al., 2016). Similarly, Ferentinos used a publicly available plant image dataset with 87,848 leaf images, splitting it into 80% for training and 20% for testing, achieving 99.53% accuracy on the test set. This study highlights the effectiveness of data partitioning and CNNs in plant species classification and disease detection (Ferentinos, 2018).

The loss function quantifies the difference between predicted and true values, minimized during training. For regression tasks, common loss functions include mean squared error (MSE), root mean squared error (RMSE), and mean absolute error (MAE), with MSE penalizing larger errors, RMSE providing interpretable

results, and MAE being more robust to outliers (Picek et al., 2022). For classification tasks, binary classification problems often use binary cross-entropy to measure the accuracy of predictions involving 'yes/no' or 'true/false' decisions (Bai et al., 2023). Custom loss functions can also be defined to suit specific project needs. For instance, Gillespie et al. developed a deep learning model called 'Deepbiosphere' and designed a sampling bias-aware binary cross-entropy loss function, which significantly improved the model's performance in monitoring changes in rare plant species (Gillespie et al., 2024).

Optimization algorithms are equally critical in DL. Common optimizers include stochastic gradient descent (SGD), adaptive moment estimation (ADAM) (Saleem et al., 2020), and RMSprop (Kanna et al., 2023). SGD updates parameters using randomly selected samples per iteration, ADAM combines momentum with adaptive learning rates, and RMSprop adjusts the learning rate of each parameter using moving averages, effectively reducing gradient oscillation (Mokhtari et al., 2023). For example, Sun et al. employed the SGD optimizer with momentum techniques in plant disease recognition tasks, which improved the convergence speed and stability of the model, thereby enhancing its accuracy (Sun et al., 2021). In a similar vein, Kavitha et al. trained six ImageNet-pretrained CNN models on an RMP dataset for rural medicinal plant classification and reported that MobileNet, optimized with SGD, achieved the best classification performance, highlighting its effectiveness in medicinal plant recognition (Kavitha et al., 2023). In another study, Labhsetwar et al. compared different optimizers for plant disease classification and found that the Adam optimizer achieved the highest validation accuracy of 98%, demonstrating its strong performance in this context. (Labhsetwar et al., 2021). Complementing these findings, Praharsha et al. evaluated multiple optimizers in CNNs and found that RMSprop, with a learning rate of 0.001 and L2 regularization of 0.0001, achieved the highest validation accuracy of 89.09%, outperforming Adam and SGD, and proving especially effective for plant pest classification tasks (Praharsha et al., 2024).

Evaluation metrics like accuracy, recall, F1 score, and area under the curve (AUC) are essential for assessing model performance, with accuracy being effective for balanced datasets but potentially misleading for imbalanced data.(Naidu et al., 2023). Recall emphasizes the model's ability to identify all relevant positive cases, essential for tasks like plant species identification, while the F1 score, the harmonic mean of precision and recall, offers a balanced evaluation, particularly for imbalanced datasets (Fourure et al., 2021). AUC evaluates model classification ability across different thresholds and is particularly useful in imbalanced classification tasks (Vakili et al., 2020). For example, Tariku et al. developed an automated plant species classification system using UAV-captured Red, Green, Blue (RGB) images and transfer learning, achieving 0.99 accuracy, precision, recall, and 0.995 F1 score, highlighting the effectiveness of these metrics in real-world tasks and the importance of recall and F1 score for handling diverse and imbalanced datasets (Tariku et al., 2023). In another study, Sa et al. introduced WeedMap, a large-scale semantic weed mapping framework using UAV-captured multispectral imagery and deep neural networks, achieving AUCs of 0.839, 0.863, and 0.782 for background, crop, and weed classes, respectively, highlighting the role of AUC in evaluating model performance across multiple categories in real-world agricultural applications (Sa et al., 2018).

Evaluation should be conducted after training and before deployment. Re-evaluation is also necessary when there are changes in data distribution or environmental conditions (Reich and Barai, 1999). For example, when encountering new plant species or ecological conditions, retraining or fine-tuning may be needed to maintain strong performance on novel inputs (Soares et al., 2017).

## 4. The applications of ML in plant research

### 4.1. Biotic and abiotic stress management

DL models are instrumental in analysing plant leaf images to detect diseases and pests, a vital component of plant protection (Shoaib et al., 2023). For instance, a 2019 study developed a back propagation neural network (BPNN) – a multilayer feedforward model trained via backpropagation and optimized through one-way ANOVA – that effectively identified rice diseases like blast and blight, highlighting its strength in pattern recognition and classification (Chaudhari & Malathi, 2023). Additionally, hyperspectral remote sensing combined with ML enables rapid detection of plant viruses like *Solanum tuberosum* virus Y, enhancing early disease identification (Polder et al., 2019). The YOLOv5s algorithm processes RGBdrone images for real-time pine wilt disease detection, suitable for large-scale monitoring (Du et al., 2024). Generative adversarial networks (GANs), consisting of a generator and a discriminator that improve through adversarial learning, have been applied in plant science for tasks such as data augmentation, plant disease detection, and growth simulation (Gandhi et al., 2018).

ML and DL advance abiotic stress management through sensors and drones for early detection, precise predictions, and improved plant resilience (Patil et al., 2024). These technologies also optimize plant stress responses, with further advancements expected in agricultural applications (Sharma et al., 2024). Hyperspectral imaging aids early disease detection related to abiotic stresses (Lowe et al., 2017). The 'ASmiR' framework predicts plant miRNAs under abiotic stresses, supporting stress-resistant crop breeding (Pradhan et al., 2023). GNN, a DL model tailored for graph-structured data, learns node representations via relationships and adjacency, and has been effectively used to predict miRNA associations with abiotic stresses by capturing complex structural patterns (Chang et al., 2024). Interested readers are encouraged to refer to the cited review, which highlights the role of bioinformatics and AI in managing abiotic stresses for food security, aiding stress gene analysis, and improving crop resilience to drought and salinity (Chang et al., 2024).

### 4.2. Plant species identification and classification

DL aids large-scale growth monitoring, identifying plant growth patterns, health, and predicting yields. An intelligent greenhouse management system uses ML and mobile networks for automated phenotypic monitoring (Rahman et al., 2024). Shapley Additive Explanations (SHAP), which quantifies each feature's contribution to model predictions, is commonly used to interpret and evaluate the performance of yield prediction models (Sun et al., 2019). Lightweight SegNet (LW-SegNet) is a CNN architecture tailored for image segmentation tasks, designed to reduce network parameters and computational demands, ensuring efficient and accurate results, particularly in resource-limited environments. Lightweight networks like LW-SegNet and Lightweight U-Net (LW-Unet) enable efficient segmentation of rice varieties in plant research (Zhang et al., 2023a, 2023b, 2023c). A hybrid model combining RF regression and radiative transfer simulation estimates wheat leaf area index (LAI) using UAV multispectral

imaging (Sahoo et al., 2023). Self-supervised Learning (SSL) trains models using patterns in unlabelled data without manual labeling, accelerating the training process; while it speeds up plant breeding with unlabelled datasets, supervised pre-training still generally outperforms SSL, particularly in tasks like leaf counting (Ogidi et al., 2023) .

CNNs enable fast and accurate plant species identification. In sustainable agriculture, a CNN-based DL model was developed to classify weeds, optimizing herbicide use for eco-friendly control (Corceiro et al., 2023). CNNs (VGG-16, GoogleNet, ResNet-50, ResNet-101) were employed to identify 23 wild grape species, showcasing the effectiveness of DL in leaf recognition and crop variety identification (Pan et al., 2024).

### 4.3. Plant growth simulation

ML explores complex molecular and cellular mechanisms in plant growth and development, such as stem cell homeostasis in arabidopsis shoot apical meristems (SAMs) (Hohm et al., 2010), leaf development (Richardson et al., 2021), and sepal giant cell development (Roeder, 2021), and the simulation of weed growth in crop fields over decades (Zhang et al., 2023a, 2023b, 2023c).

Various algorithms have been applied to study plant development. For example, image processing and ML using SVM and RF were used to analyze Cannabis sativa callus morphology, with SVM showing higher accuracy, while genetic algorithms optimized PGR concentrations to validate the model (Hesami and Jones, 2021). A microfluidic chip was created to simulate pollen tube growth, and the 'Physical microenvironment Assay (SPA)' method was established to study mechanical signal transmission during pollen tube penetration of pistil tissues (Zhou et al., 2023).

Many groups are utilizing the latest computational technologies and algorithms to develop tools that enhance the efficiency and precision of plant biology research (Muller & Martre, 2019). The virtual plant tissue (VPTissue) software simulates plant developmental processes, facilitating the integration of functional modules and cross-model coupling to efficiently simulate cellular-level plant growth (De Vos et al., 2017). ADAM-Plant software uses stochastic techniques to simulate breeding plans for self- and cross-pollinated crops, tracking genetic changes across scenarios and supporting diverse population structures, genomic models, and selection strategies for optimized breeding design (Liu et al., 2019). The L-Py framework, a Python-based L-system simulation tool, simplifies plant architecture simulation and analysis, with dynamic features that enhance programming flexibility, making plant growth model development more convenient (Boudon et al., 2012). Many groups are also developing advanced computational tools to accurately simulate plant morphological changes at various stages of growth (Boudon et al., 2015). A 3D maize canopy model was created using a t-distribution for the initial model, treating the maize whorl – leaves, stem segments, and buds – as an agent to precisely simulate the canopy's spatial dynamics and structure (Wu et al., 2024).

Significant advancements from 2012 to 2023 have enhanced our understanding of plant biology. In 2012, researchers used live imaging combined with computational analysis to monitor cellular and tissue dynamics in *A. thaliana* (Cunha et al., 2012). The introduction of the Cellzilla platform in 2013 enabled simulation of plant tissue growth at the cellular level (Shapiro et al., 2013). A pivotal study in 2014 focused on the WOX5-IAA17 feedback loop, which is essential for maintaining the auxin gradient in *A. thaliana* (Tian et al., 2014). By 2016, research explored plant signaling pathways and mechanical models to analyze sepal growth and morphology (Hervieux et al., 2016). In 2018, studies delineated the expression pattern of the *CLV3* gene in SAMs (Zhou et al., 2018a, 2018b), followed by 2019 research on leaf development and chloroplast ultrastructure (Kierzkowski et al., 2019), and the *TCX2* gene's role in maintaining stem cell identity (Clark et al., 2019). Research in 2020 focused on epidermis-specific transcription factors affecting stem cell niches (Han et al., 2020), and 2021 introduced new modeling techniques for root tip growth and stem cell division (Marconi et al., 2021). In 2022, 3D bioprinting was used to study cellular dynamics in both *A. thaliana* and *Glycine max* (Van den Broeck et al., 2022). The latest studies in 2023 provided new insights into weed evolution and applied advanced DL techniques for plant cell analysis (Feng et al., 2023). These milestones demonstrate the integration of computational tools and empirical datasets in plant science, enabling innovative methods and applications that propel the field forward.

### 4.4. Plant cell segmentation

Accurate cell segmentation is crucial for understanding plant cell morphology, developmental processes, and tissue organization. Recent advancements in DL and computer vision have led to the development of various specialized tools for segmenting plant cell structures from complex microscopy data. This section provides an overview of key tools, highlighting their core methodologies, applications, and advantages in plant research.

PlantSeg is a neural network-based tool designed for high-resolution plant cell segmentation. It starts with image preprocessing, including scaling and normalization, followed by U-Net-based boundary prediction to identify cell boundaries (Wei et al., 2024). The boundary map is transformed into a region adjacency graph (RAG), where nodes represent image regions and edges represent boundary predictions. Graph segmentation algorithms, such as Multicut or GASP, partition the graph into individual cells, and post-processing ensures the segmentations align with the original resolution and corrects over-segmentation (Wolny et al., 2020). By streamlining these processes, PlantSeg supports high-throughput analysis of plant cell dynamics, particularly for confocal and light sheet microscopy data (Vijayan et al., 2024). This tool not only improves segmentation efficiency but also handles large-scale datasets, providing robust support for long-term monitoring of plant cell behavior.

Complementing PlantSeg, the Soybean-MVS dataset leverages multi-view stereo (MVS) technology to provide a 3D imaging resource, capturing the full growth cycle of soybeans and enabling precise 3D segmentation of plant organs (Sun et al., 2023). This dataset plays a significant role in plant growth and developmental research, offering fine-grained data support for dynamic analysis of long-term growth processes.

Other tools focus on generalizability and adaptability across various plant species and imaging modalities. Cellpose utilizes a convolutional neural network (CNN), which performs well in segmenting different cell types and shapes, especially in dynamic plant structures and large-scale image analysis, improving scalability (Stringer et al., 2021). This feature enables Cellpose to maintain high accuracy and robustness under diverse experimental conditions when processing plant cells.

DeepCell uses CNNs for plant cell segmentation and supports cell tracking and morphology analysis, offering robust tools

for phenotype research. It excels in handling complex cellular dynamics and large-scale datasets, making it ideal for long-term monitoring of plant phenotypes (Greenwald et al., 2022).

Ilastik provides an interactive ML-based segmentation approach, combining flexibility and accuracy. Its user-guided training enables adaptation to diverse plant datasets and experimental conditions, making it valuable for cross-species and multi-modal plant data analysis (Robinson & Vink, 2024).

Finally, MGX (MorphoGraphX) specializes in 3D morphological analysis of plant tissues by processing 3D microscopy data to visualize and quantify cell shapes, sizes, and spatial patterns. It supports studies on cell interactions and tissue growth, offering precise tools for plant tissue development research (Kerstens et al., 2020).

In conclusion, despite differences in algorithms and interfaces, these tools collectively advance plant microscopy by minimizing manual segmentation, enhancing reproducibility, and enabling high-throughput and multidimensional analysis. Their complementary strengths offer researchers diverse options, from 2D segmentation to full 3D tissue modeling, tailored to specific experimental needs, and significantly improve efficiency and precision in plant cell and tissue analysis.

## 5. Summary

ML and image recognition technologies show great promise in plant science, yet several challenges must be addressed for their effective application (Xiong et al., 2021). Figure 2 illustrates the

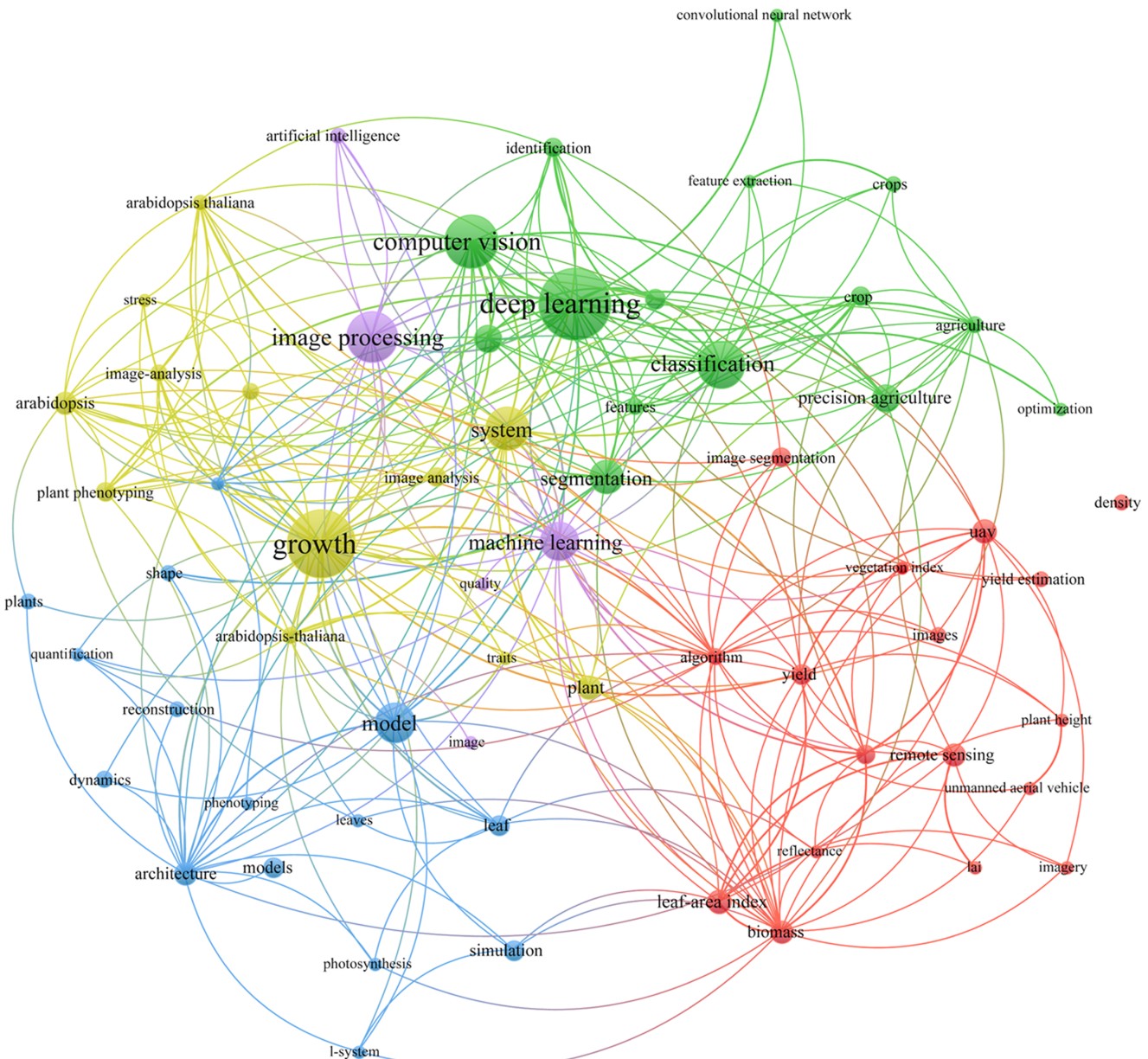

**Figure 2.** A keyword network analysis of DL in plants. A keyword analysis of plant AI technologies reveals clear technological connections. Blue lines indicate image processing technologies, and green lines represent plant phenotyping and growth analysis. At its core, 'DL' links to 'ML', 'image processing,' and 'computer vision.' Technologies such as 'remote sensing' and 'precision agriculture'. The relationships between terms like 'plant growth', 'diseases', 'phenomics', and 'smart agriculture' indicate the growing integration of AI and ML in improving plant practices.

**Table 3.** Challenges and future trends of ML and image recognition technologies in plant science

|  | Category | Content | References |
|---|---|---|---|
| Challenges | Image quality and diversity | Imaging variations, such as lighting, angle, and background, can significantly affect recognition accuracy. | Shoaib et al. (2023) |
|  | Inter-class similarity | The similarity in appearance among different plant species can present challenges to accurate classification. | Jeyapoornima et al. (2023) |
|  | Intra-class variation | Individuals of the same plant species can show significant variations in appearance, including growth stages and seasonal changes. | Harris (1913) |
|  | Insufficient data | Limited image data for rare or endangered plants makes training effective ML models challenging. | Cong and Zhou (2023) |
|  | Complexity of background | Plant images often have complex backgrounds, such as soil and other plants, which can interfere with recognition and feature extraction. | Wang et al. (2008) |
|  | Real-time processing requirements | Real-time plant identification on mobile devices faces challenges due to high demands on processing speed and resource consumption. | Padhiary et al. (2023) |
|  | The multi-label classification problem | An individual plant image may require multiple labels, like species and disease type, increasing classification complexity. | Anh et al. (2022) |
|  | Adaptability and scalability | Identification systems need good adaptability and scalability to accommodate new species discoveries and updated classification standards. | Rao et al. (2022) |
| Future trends | Further applications of DL | DL, especially CNNs, is effective for image recognition and has potential for optimizing plant sample identification and categorization. | Chen et al. (2023) |
|  | The application of weakly supervised learning and unsupervised learning | Due to high annotation costs, weakly supervised and unsupervised learning methods using unlabelled data are expected to be more widely adopted as cost-effective solutions. | Adke et al. (2022) |
|  | The development of fine-grained image recognition | Fine-grained image recognition targets distinguishing highly similar species, with future research addressing high inter-class similarity. | Šulc and Matas (2017) |
|  | The exploration of cross-domain learning techniques | Transferring advanced technologies from other domains to plant image recognition may help overcome specific challenges. | Chulif et al. (2023) |
|  | Increasing interpretability and transparency | As ML models gain prominence in plant science, their interpretability and transparency are key to understanding decision-making processes. | Paudel et al. (2023) |
|  | The usage of mobile devices and edge computing | Mobile devices and edge computing for image capture and initial processing will enable real-time plant recognition and data collection in the field. | Khan et al. (2023) |
|  | The application of multimodal learning | The system's accuracy and robustness can be improved by integrating image data with genetic and ecological information through multimodal learning. | Zhou et al. (2021) |
|  | The integration of cloud computing and big data technologies | Cloud computing and big data will increasingly manage large plant image datasets, offering improved computational resources and storage. | Singh (2018) |
|  | Sustainability and environmental monitoring applications | ML and image recognition will support plant conservation, species monitoring, and environmental assessments, aiding sustainable development goals. | Wongchai et al. (2022) |

keyword network analysis of DL in plant research publications, while Table 3 outlines the challenges and future trends in ML and image recognition technologies within plant science.

In summary, we examined the usage of DL-based image recognition in plant science, covering plant feature extraction, classification, disease detection, and growth analysis. We highlight the importance of data acquisition and preprocessing methods like high-resolution imaging, drone photography, and 3D scanning, as well as techniques for improving data quality. Various feature extraction methods – such as colour histograms, shape descriptors, and texture features – are reviewed for plant identification. The development of ML models, especially CNNs, is also discussed, alongside current challenges and future prospects. Despite progress, challenges remain. Future research should aim to apply methods across diverse plant systems, refine data acquisition, and enhance algorithm efficiency. Advancements will likely improve model generalization and interpretability, with interdisciplinary collaboration in plant biology, mathematics, and computer science being crucial to addressing upcoming challenges.

**Open peer review.** To view the open peer review materials for this article, please visit http://doi.org/10.1017/qpb.2025.10018.

## Acknowledgements

We acknowledge Manjun Shang for critical reading our manuscript. We apologized to colleagues whose work was not included or described in this review due to limited space.

**Competing interest.** None.

**Data availability statement.** No data and coding involved in this manuscript.

**Author contributions.** KYH, HH and YZ conceived the study. KYH wrote the manuscript. HH and YZ revised the manuscript.

**Funding statement.** This work was supported by National Natural Science Foundation of China (No. 32202496 to H. H.), Nanjing Forestry University (start-up funding to H.H.), and Jiangsu Sheng Tepin Jiaoshou Program (to H.H.).

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
