## [Reviewer Report]

This review describes how advances in machine learning and image processing can be applied to plant science. It describes major trends and methods to use these novel methods in the field of plant biology. It systematically illustrates different methods to acquire data, preprocess it, and to extract features from it. It then continues to mention different types of machine learning algorithms and how to train them. At last, the authors describe previous important work in the field and describe future trends and challenges.

Overall, I believe that this is an important manuscript, that explains the trends of machine learning to plant biologists. As such, I believe that this manuscript needs to better explain the technical terms and when generally it is better to use one option over the other, in a language that is understandable to a trained biologist. I also think that it is important to further stress what are the specific challenges and points when comparing application of machine learning to plant science. Moreover, I found several cases in which the work described in the manuscript does not match the information supplied in the cited work.

Major points:

- It is worth to explain in the introduction that the usage of machine learning in the plant biology field lags behind the state-of-the-art of neural networks. For example, these days language models are revolutionizing the field of A.I. and we still don’t see a major use for these methods in plant biology.

- Terms should be better defined: Image classification, Image recognition, object detection are somewhat similar tasks but have distinct precise definitions and purposes. I think that when using such a term, a clear definition and explanation about the task that is performed by the algorithm should be added, and terms should be kept – FRCNN uses the term object detection to describe their work.

- I think that for this review, the authors should emphasize more what is special about plants. For example, which preprocessing, feature extraction, data augmentation are in general used in machine learning, and how are they better suited for plant research and why, and maybe even add these points to the tables.

- In table 2, the techniques suggested are usually not labor intensive, and it is important to stress that: a. these methods also lead to information loss, and thus there is a need to find the right balance between simplicity of the model, and loss of potentially important information. b. that the labor-intensive part is the actual labeling if the training data, and in many cases that is the bottle neck that can cause such projects to fail.

- In the computer vision field, the labeled data is often referred to as ground truth, and it is important to introduce such common-used terms in the review and explain them.

- Regarding table 3: I believe that the trend in computer vision is not to hand-craft feature extraction but allow the algorithm to learn which features to extract by itself. Also, the separation between tables 2 and 3 is somewhat artificial. I’m not sure whether this table is needed.

- Table 4: can the manuscript include some general ranking of performances of the different algorithms to perform a task (object detection- for example)? For someone, not familiar with the field this table can be overwhelming.

- Figure 1: This figure is hard to comprehend, processes seem to be happening in parallel. A more linear layout of the algorithm would be very beneficial.

- In lines 216-217, there is an explanation of data partitioning, I think that the practical use of each set is missing from the description. When does one use the training set? – for every image used while training, validation – once per epoch, or a way to evaluate performance mid training on unseen data, and test – to compare between models, and evaluate performance at the end of training. The way it is currently written is unclear for someone that is not from the field. What is the difference between a parameter and a hyper-parameter?

- Evaluation metrics are mentioned but not explained. Each should include what is it measuring, why, and what it is detecting strongly compared to the other metrics.

- What are GANs? (explain and cite).

- What is a BP NN, YOLOv5s? What is different between these NNs and a regular CNN?

- Session 5.2.1 – you didn’t mention genomes previously at all. What differences are there between image and sequence ML? With which data? What about language models?

- I think that the NN plantseg that segments cells in plants is important to cite and explain as a different use case of image processing to analyze microscopy data (https://doi.org/10.7554/eLife.57613).

Minor points:

- Authors should ensure that they accurately explain work cited. For example: in line 54, Ullah et al. 2024 does not seem to state the number 83% in their paper.

- Line 54: citation missing for Mhango and colleagues

- In the data acquisition, I think it make sense to also mention the use of regular cameras and phone, which have lower image quality and precision, but easier to obtain large amounts of data.

- In line 106: There is a need to explain what data augmentation actually means in practice: one, performs operation on original images, feed those to the algorithm that sees them as new data sets. This allows for higher richness and robustness in training.

- In line 135: the word “with” should be replace with “from”.

- In line 186: can the authors provide a rough estimate for an acceptable size of a dataset to yield good results?

- Line 191: this paragraph starts with “In summary” yet when the paragraph ends the section continues.

- Line 194: change the word “field” with “plant biology”

- Loss function -can be also customized and designed by oneself, according to the needs of the project. The importance and role of the loss function should be further stressed.

- Line 237: citation in wrong format

- Line 238: please add a sentence about when one should evaluate their model.

- Line 380: what is SHAP?

- Line 385: 98% accuracy in what? What was the task at hand?

- Line 385: the authors start the sentence with Sun et al. but cited Zhang et al.

- The plant Village dataset, has the potential to be an important resource, it should be mentioned sooner in the paper, and stress its importance and other datasets like it.

- Line 426: Table 5, should be Table 6.

---

## [Reviewer Report]

This is a review paper summarizing image processing and machine learning in plant science. A review paper is supposed to provide information that is well defined and clearly summarized so the readers can form an accurate knowledge base regarding a certain topic. But this paper is not well written, lacking effective summary and synthesis of the literature. The overall issue is that the title is too broad and limited effort on synthesizing and summarizing the literature. Suggest the authors to rename the title and narrow down the focus area and restructure the outline.

Specifically, the authors piled multiple papers in tables and just simply describe what was done in each paper. Suggest the authors to digest the content and summarize them better. And there are an inefficient number of papers included for a particular topic.

For example, the data acquisition section (Table 1), the category for the techniques is lack of consideration. The categories should be clearly divided into either the type of camera (visual, multispectral, lidar), the type of platform (UAV or indoor settings?), or the type of applications (plant morphology, plant physiology, growth, yield, etc.)? Does 3D scanning technology equate to LiDAR? “environmental monitoring sensor” is also included and it is not even images dataset.

The authors are suggested to clearly define different types of algorithms at a higher level, difference among commonly used terms AI, machine learning, and deep learning. You have combined both image data and “spreadsheet” data which may require different types of algorithms to handle. The applications included in this review are also limited. You included disease, but what about abiotic stress? Most of the studies are agricultural related studies (crop science) and less with plant science.

---

## [Reviewer Report]

The review is now much better explained than before, and it is much more comprehensive.

Yet there is still some disconnect between the ML and plant sciences parts.

Even though the paper is much better written, there are still some points to address:

Major comments:

1. The explanations about the algorithms are much more straightforward now and appear much more didactic. Yet, in many cases (for example, in sections 3.1 and 3.3) – a connection to the actual plant science is missing. I think that a small addition to each of the algorithms about how they do on a plant-relevant study and what the apparent citation managed to do with it will add significant value (like in line 229). If no work related to plants exists – explain the need and potential for such a work or remove it entirely. Maybe the authors could combine sections 3 and 4?

2. Some errors occurred; Figure 2 is not seen in the final PDF.

3. Section 4.4 lacks flow. It lacks an introduction and summary paragraphs; it’s just a list of algorithm descriptions. Each section should have more insight than a list of existing manuscripts or tools.

4. The addition of the performance score is useful; a numerical score on known metrics would be better, even though I recognize that it can be challenging to curate such values due to the use of different metrics in different manuscripts. If it’s impossible, can the method by which the qualitative descriptions of performances (high/ medium, etc.) were obtained be described?

Minor comments:

- Fig. 1 seems to have a black background with large white boxes. This may be due to my computer’s configuration, but in any case, please make sure that the background stays white across all platforms.

- Maybe combine the very short section 5 with section 6?

Line 74. The idea that performance has been tested misses a conclusion - How did the CNNs perform on the public databases? Were they any good?

Line 84. What are some of the new insights that were revealed by LLMs?

Line 115. Maybe it is necessary to say that CNNs can be fed with the raw images and learn the features by themselves.

Line 119. Who added the physiological features? Were they manually analyzed and added to the algorithm as metadata? Or maybe they were automatically computed?

Lines 135 - 144. The paragraph is a bit repetitive. The same sentence appears several times with slightly different wording. Consider condensing it.

---

## [Reviewer Report]

The paper makes a valuable contribution by compiling tools and techniques used in plant image processing. It fits the journal’s scope, is timely, and is useful for both new and experienced researchers in the field. However, the authors should add more critical comparisons between methods to improve clarity and depth. For example, while many models and techniques are listed, the paper lacks critical insight or comparison — e.g., which methods are best for which tasks?

---

## [Editor Report]

Thank you very much for the time and effort you’ve dedicated to addressing the comments. The revised version looks excellent and represents a strong contribution. If possible, just a few minor remarks remain to be addressed to further strengthen the work.